# Deciphering the Effect of Microbead Size Distribution on the Kinetics of Heterogeneous Biocatalysts through Single-Particle Analysis Based on Fluorescence Microscopy

**Emilio Muñoz-Morales [1], Susana Velasco-Lozano [1], Ana I. Benítez-Mateos [2], María J. Marín [3], Pedro Ramos-Cabrer [4,5] and Fernando López-Gallego [1,2,5,6,*]**

[1] Heterogeneous Biocatalysis Laboratory, Instituto de Síntesis Química y Catálisis Homogénea (iSQCH), CSIC-Universidad de Zaragoza, C/Pedro Cerbuna 12, 50009 Zaragoza, Spain; emilio.morales193@gmail.com (E.M.-M.); svelasco@unizar.es (S.V.-L.)

[2] Heterogeneous Biocatalysis Laboratory, CICbiomaGUNE, Edificio Empresarial "C", Paseo de Miramón, 182, 20014 Donostia-San Sebastián, Spain; ana.benitez@dcb.unibe.ch

[3] Servicio General de Apoyo a la Investigación—SAI, Servicio de Microscopía Electrónica de Sistemas Biológicos, Universidad de Zaragoza, C/Pedro Cerbuna 12, 50009 Zaragoza, Spain; mjmarin@unizar.es

[4] Magnetic Resonance Imaging Lab, CICbiomaGUNE, Edificio Empresarial "C", Paseo de Miramón, 182, 20014 Donostia-San Sebastián, Spain; pramos@cicbiomagune.es

[5] IKERBASQUE, Basque Foundation for Science, Maria Diaz de Haro 3, 48013 Bilbao, Spain

[6] ARAID, Aragon foundation for Science, 50009 Zaragoza, Spain

* Correspondence: flopez@cicbiomagune.es

**Abstract:** Understanding the functionality of immobilized enzymes with spatiotemporal resolution and under operando conditions is an unmet need in applied biocatalysis, as well as priceless information to guide the optimization of heterogeneous biocatalysts for industrial purposes. Unfortunately, enzyme immobilization still relies on trial-and-error approximations that prevail over rational designs. Hence, a modern fabrication process to achieve efficient and robust heterogeneous biocatalysts demands comprehensive characterization techniques to track and understand the immobilization process at the protein–material interface. Recently, our group has developed a new generation of self-sufficient heterogeneous biocatalysts based on alcohol dehydrogenases co-immobilized with nicotinamide cofactors on agarose porous microbeads. Harnessing the autofluorescence of $NAD^+(P)H$ and using time-lapse fluorescence microscopy, enzyme activity toward the redox cofactors can be monitored inside the beads. To analyze these data, herein we present an image analytical tool to quantify the apparent Michaelis–Menten parameters of alcohol dehydrogenases co-immobilized with $NAD(P)^+/H$ at the single-particle level. Using this tool, we found a strong negative correlation between the apparent catalytic performance of the immobilized enzymes and the bead radius when using exogenous bulky substrates in reduction reactions. Therefore, applying image analytics routines to microscopy studies, we can directly unravel the functional heterogeneity of different heterogeneous biocatalyst samples tested under different reaction conditions.

**Keywords:** protein immobilization; alcohol dehydrogenase; NAD(P)H; biocatalysis; agarose; bio-redox

## 1. Introduction

Heterogeneous biocatalysis is an attractive approach to perform more efficient, robust, and sustainable chemical processes [1]. For this reason, enzyme technologists are encouraged to develop

highly active and stable heterogeneous biocatalysts [2]; however, molecular characterization of the supported enzymes is rather limited. Thus far, techniques for the molecular characterization of solid-supported enzymes are scarce, unlike chemical catalysis, where molecular characterization drives the design and optimization of heterogeneous catalysts [3,4]. Recently, several authors have deeply reviewed different techniques for the advanced characterization of heterogeneous biocatalysts [5–7]. They summarized an analytical toolbox that provides valuable information about the function, the structure, and the dynamics of the immobilized proteins. Many of these techniques rely on the recent advances in fluorescence microscope applied for fundamental biological studies that inform about the spatial, dynamic, and structural organization of proteins across mimetic biostructures [8–12]. The vast majority of these studies have been carried out over model and uniform solid materials based on inorganic wafers functionalized with self-assembly monolayers [13]. Unfortunately, these systems are far from the architectures exploited for the fabrication of heterogeneous biocatalysts with industrial purposes [14]. Mostly, the commercially available carriers used for enzyme immobilization are microbeads (silica, biopolymers, organic polymers … ) with a significant polydispersity in particle size (Table 1).

**Table 1.** Bead size distribution of commercial carriers for protein immobilization.

| Carrier | Bead Size (μm) | | | | Refs |
|---------|----------------|---|---|---|------|
| | *Standard* | *Large* | *Small* | *Wide Range* | |
| ABT$^{TM}$ | 50–150 | n.a | 20–50 | n.a | [15] |
| Purolite$^{TM}$ | 150–300 | 300–710 | n.a | 300–1200 | [16] |
| Relizyme$^{TM}$ | 100–300 | 200–500 | n.a | n.a | [17] |
| CPG-Silica | 120–200 | n.a | 20–80 | n.a | [18] |
| EziG$^{TM}$ | 75–125 | n.a | n.a | n.a | [19] |

The data have been obtained from the company's' websites. For more information, see the references. n.a: not available.

Table 1 shows the size range of the most commercialized carries for enzyme immobilization in industrial biocatalysis. The range of the bead size in one sample may vary from the 50 μm reported for EziG$^{TM}$ to the 900 μm reported for Purolite$^{TM}$. When enzymes are immobilized on these carriers, their functional and structural characterization relies on macroscopic studies based on bulk experiments that assume that all the particles (beads) in the same sample are equal. Unfortunately, these studies mask the functional differences between populations of beads with different sizes. To unveil bead-to-bead functional variability and thus study the effect of the size dispersion on the functional heterogeneity of one sample, single-bead analyses with spatiotemporal resolution are needed.

Beside the nature of the carriers (i.e., hydrophobicity, porosity, reactivity, etc.,), it is very well known that the carrier particle size has an effect on the effectiveness of the immobilized enzymes. The group of Prof. Illanes has extensively studied how the bead size affects the internal diffusion restrictions of substrates and consequently the catalytic performance of different hydrolases immobilized on agarose microbeads activated with aldehyde groups [20–22]. They proved that the enzymes immobilized on carriers with a large mean particle size exhibit lower apparent catalytic efficiency than samples with a smaller mean particle size. This effect was even more dramatic using high loadings of immobilized enzymes. Similar evidences were reported using oxygen-dependent flavin oxidases co-immobilized with chemical oxygen sensors [23]. Although these results are quantitative and meaningful, they do not consider the size dispersion of the carrier, which likely affects the bead-to-bead functionality within the same sample. More recently, Consolati et al. have developed a single particle analysis to monitor the intraparticle pH gradients created by the action of penicillin G acylase co-immobilized with the yellow fluorescence protein as a pH biosensor [24]. They qualitatively studied the microscopic data, but did not quantitatively link the pH gradients to the enzyme kinetics.

Inspired by single-cell studies and using some of the tools developed for that purpose [25,26], we have herein designed and developed a platform for image analytics of heterogeneous biocatalysts. The image processing of time-lapse fluorescence microscopy experiments linked to mathematical modeling enables us to estimate apparent kinetic parameters at both single particle and sub-micrometric levels. These analyses were tested and validated using a new generation of self-sufficient heterogeneous biocatalysts recently developed in our group [27,28]. In these systems, the NAD(P)H-dependent alcohol dehydrogenases are co-immobilized with their corresponding cofactor on porous agarose microbeads. Within the pores, the enzymes are tightly bound to the carrier surface, while the redox cofactors shuttle from one active site to the other without diffusing out to the reaction bulk. The intraporal traveling of cofactors relies on an association/dissociation equilibrium established by the ionic interactions between the NAD(P)H phosphate groups and the positive charges (mainly amine groups) of the carrier surface. In this work, we have developed a new analytical tool to understand the functional variability of enzymes at the protein–solid interface with spatiotemporal resolution. By monitoring the intraparticle autofluorescence of reduced cofactors under operando conditions, we deciphered a bead-to-bead functional variability associated to the size dispersion of the carrier where the enzymes were immobilized. These data confirm the macroscopic studies [20–22,29] that demonstrate the effect of the particle size on the catalytic effectiveness. Therefore, we have been able to quantitatively characterize the kinetics of "ready-to-use" alcohol dehydrogenases co-immobilized with their corresponding cofactors. This architecture is enormously interesting to enhance the cost-efficiency of the process, since exogenous cofactors are no longer required.

## 2. Results and Discussion

### 2.1. Validation of Image Analytics

Lastly, our group has developed a time-lapse fluorescence microscopy methodology to study the activity of NAD(P)H-dependent alcohol dehydrogenases co-immobilized with their corresponding redox cofactors on porous microbeads [27,28]. Until now, we manually analyze this information bead by bead, suffering user bias in the final results. To automate this process and set automatic thresholds, we have developed an innovative work-flow under the Image J environment to select the contours of the beads as regions of interest (ROIs) in all the frames of one temporal image stack. To test this work-flow, we selected two systems previously developed in our lab (Scheme 1).

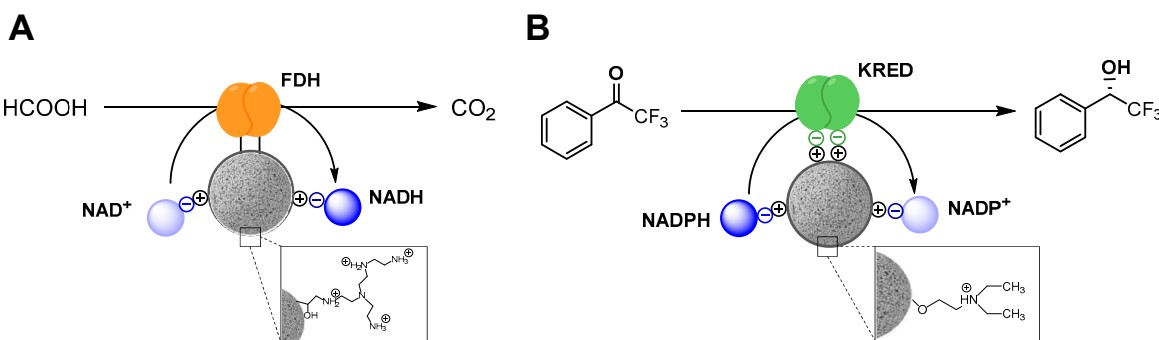

**Scheme 1.** Self-sufficient heterogeneous biocatalysts. (**A**) Formate dehydrogenase from *Candida boidinii* (FDH) co-immobilized with nicotinamide-adenine-dinucleotide sodium salt (NAD$^+$) on agarose microbeads activated with polyethyleneimine (PEI). The reaction is triggered by adding exogenous formic acid. (**B**) Ketoreductase P1-A04 (Codexis$^{®}$) (KRED) co-immobilized with NADPH on agarose microbeads activated with diethylaminoethyl (DEAE). The reaction was triggered with exogenous 2,2,2-trifluoacetophenone (TFA).

In the first system, formate dehydrogenase from *Candida boidinii* (FDH) was irreversibly immobilized on porous agarose microbeads coated with polyethyleneimine (PEI), while NAD$^+$

was ionically adsorbed on that cationic polymeric bed (Scheme 1A). Under operando conditions, we can monitor the reaction progress within each bead through the increasing of NADH autofluorescence. Here, the immobilized FDH reduced the immobilized $NAD^+$ to NADH in the presence of exogenous formic acid (left panel Figure 1A). In the second system, Ketoreductase P1-A04 (Codexis®) (KRED) was co-immobilized with NADPH on porous agarose microbeads activated with tertiary amine groups through ionic exchange (Scheme 1B). The single-particle microscopy studies show that the internal bead intensity decreases along the time (left panel, Figure 1B) when the immobilized NADPH was oxidized to $NADP^+$ by KRED in the presence of 2,2,2-trifluoroacetophenone (TFA). In both types of samples, this image tool also provides numerical data that can be plotted as the average fluorescence intensity ($I_f$) of each bead versus time (middle panels, Figure 1). Then, the PCAT tool (MATLAB script) reported by Bäuerle et al. [30] for single-cell enzymatic analysis was used to estimate the kinetic parameters of immobilized enzymes on single beads. To implement a fully analytical method that calculates the apparent Michaelis–Menten (M-M) constants of the supported enzymes, we normalized the time courses by transforming the raw data ($I_f$) into progress curves where arbitrary units of product concentration ($PU \times \mu m^{-3}$) increase along the time (right panels, Figure 1) (Section 3.6). This volumetric normalization is possible due to the uniform distribution of cofactors across the microstructure of the beads, and is needed to compare the results between beads with different sizes (Figure S6).

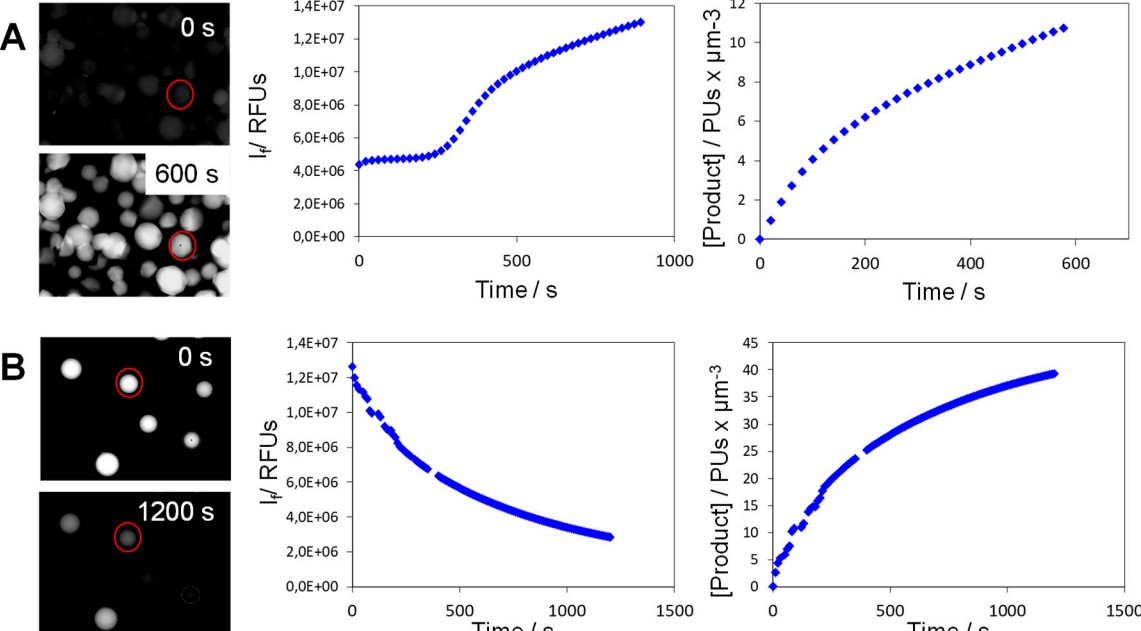

**Figure 1.** (**A**) Single-particle time courses extracted from microcopy images of agarose microbeads co-immobilizing formate dehydrogenase and $NAD^+$. The reaction was triggered with 10 mM formic acid. (**B**) Single-particle time courses extracted from microscopy images of agarose microbeads co-immobilizing KRED and NADPH. The reaction was triggered with 10 mM of TFA. Left: microscopy images of initial and final times. Center: time-course of increasing (**A**) or decreasing (**B**) fluorescence intensity directly extracted from the time-frames of one bead-the region of interest (ROI). Right: Normalized progress curves of product volumetric concentration along the reaction time for the selected ROI (right micrographs). These plots are derived from the central plots that contain the raw data acquired from the microscope.

These normalized data (as.csv format, right panels of Figure 1) serve as input files to calculate the M-M constants of the immobilized enzymes using the PCAT tool that fits the experimental data to the Lambert W function of the M-M equation through an analytical model, see Section 3.8 (Equation (6)). For each selected ROI, we obtained one plot that overlaps fitting curves and normalized data (Figure 2 top panel). The model estimates the following parameters (Figure 2, bottom panel); the maximum

concentration of formed product ($[P_f]$) and the apparent M-M kinetic constants of the immobilized enzymes toward their corresponding cofactor. These parameters are: maximum rate ($V_M$), M-M constant ($K_M$), and catalytic performance ($V_M/K_M$). Moreover, fitting the linear part of the progress curve (product concentration below 20% of $[P_f]$), we calculated the initial rate within the selected ROI (see Section 3.9). The accuracy of all these kinetic parameters was assessed calculating the residual values for the nonlinear analytical fitting of Lambert W function and $R^2$ for the linear regressions used to calculate $V_o$. Residuals is a scalar indicator that means the sum of squares of each time point. Selected fitting presented a square root of the residuals that is lower than 15% of the [Pf] value (Figure S1). Figure 2 shows the estimated values of apparent M-M parameters toward immobilized nicotinamide cofactors using the normalized progress curves presented in Figure 1 for two types of enzymatic redox reaction: $NAD^+$ reduction (Figure 2A) and NADPH oxidation (Figure 2B). The M-M parameters herein estimated are considered as apparent ones, since they account for both the external and internal diffusion restrictions suffered by the substrates to reach the active sites of the immobilized enzymes confined in a porous environment. Thus, these apparent parameters [31] are affected by all the parameters that affect substrate diffusivity.

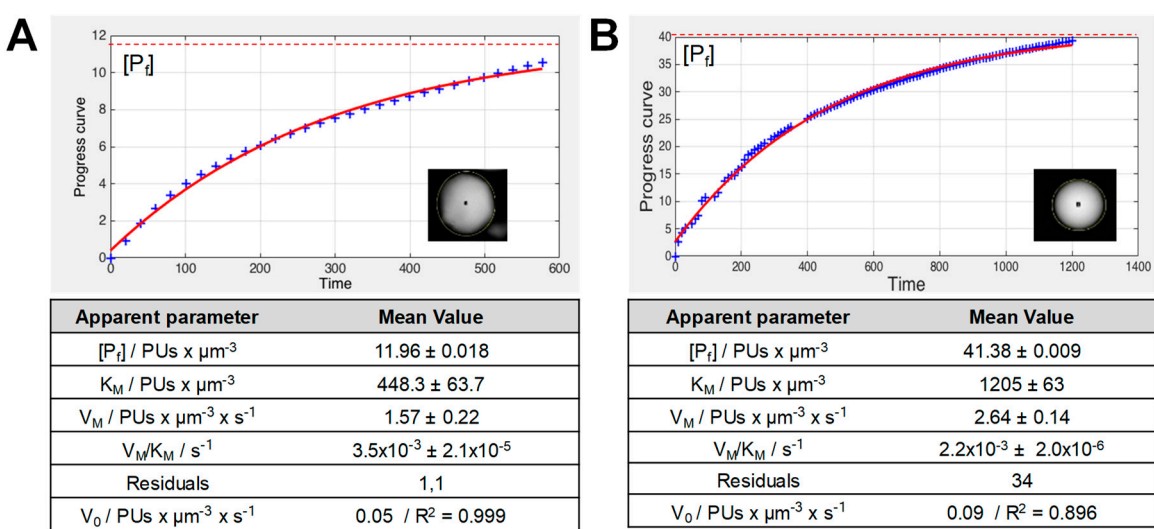

**Figure 2.** PCAT fitting of time-course data acquired from single-particle analysis shown in Figure 1A (**A**) and in Figure 1B (**B**). The fitting done through an analytical approximation gives rise to the following apparent kinetic parameters (bottom tables): $[P_f]$ = maximum product concentration (fluorescence intensity (RFU) x $\mu m^{-3}$) achieved during the reaction course (red dash line). $K_M$: Michaelis–Menten constant towards the redox cofactor expressed as arbitrary concentration of substrate (RFU × $\mu m^{-3}$). $V_M$: Maximum enzyme rate for the product generation (RFU x $\mu m^{-3}$ × $min^{-1}$. The confidence of the PCAT fitting was assessed by the residuals plots (Figure S1). The mean value of each parameter was determined from three iterations of the PCAT tool using the analytical method with different input $K_M$ and $V_M$ values. For plot A; Set 1: $K_M$ (i) = 1/$V_M$ (i) = 0.01; Set 2: $K_M$ (i) = 10/$V_M$ (i) = 0.1; Set 3: $K_M$ (i) = 100/$V_M$ (i) =1. For plot B; Set 1: $K_M$ (i) = 5/$V_M$ (i) = 0.05; Set 2: $K_M$ (i) = 10/$V_M$ (i) = 0.1; Set 3: $K_M$ = 30/$V_M$ = 0.3). $V_0$: Initial rate of the enzyme reaction. This parameter was determined through fitting the data of product concentration versus time using the equation ($[P] = V_0 \times t$). The data used for this linear regression were those whose RFU × $\mu m^{-3}$ were lower than 20% of the value of $[P_f]$.

## 2.2. Analysis of the Functional Heterogeneity within Samples of Immobilized Enzymes

To shine light on the sample heterogeneity, we have exploited a spatiotemporal analysis of single particles to study the effect of the radius size on the apparent kinetic parameters of immobilized enzymes. To that aim, we plotted the apparent kinetic parameters for individual beads versus their radius size (Figures 3 and 4). In the case of FDH co-immobilized with $NAD^+$, the bead size negligibly affects the apparent kinetic parameters of the immobilized enzyme in the presence of formic acid (Figure 3).

Although both $V_0$ and $V_M$ showed a slight negative correlation with the radius size (Figure 3A,B), the rate differences between beads were not significant at all to impact on catalytic performance.

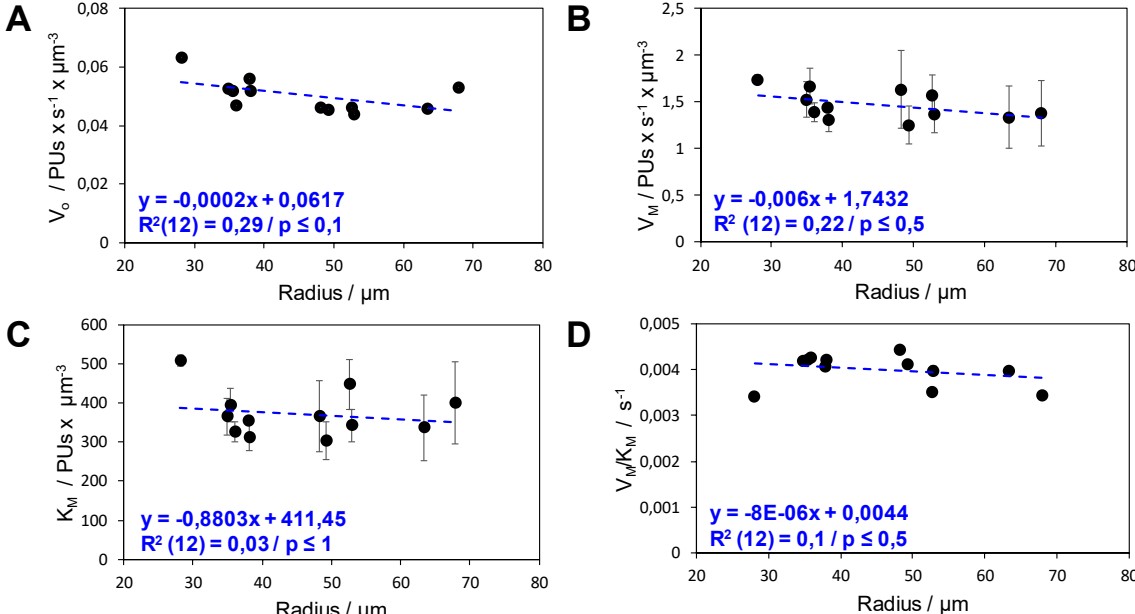

**Figure 3.** Effect of the particle radius on the kinetic parameters of the immobilized FDH and NAD$^+$ using formic acid as substrate. For single particle analysis, apparent $V_0$ (**A**), $V_M$ (**B**), $K_M$ (**C**), and $V_M/K_M$ (**D**) were determined and plotted against the radius of each particle (circles). Errors for apparent $V_M$ and $K_M$ were calculated from three different iterations of PCAT using a different input data set. The dashed blue line represents the linear correlation between each kinetic parameter and the particle radius. Linear regression equations are shown within each graph. The multiple correlation coefficient $R^2$ and the *p* were calculated with an ANOVA statistical analysis. The number of samples is indicated in brackets beside the $R^2$ value.

On the contrary, the single-particle image analysis of KRED co-immobilized with NADPH using 2,2,2-trifluoacetophenone (TFA) as substrate presented a much stronger negative correlation between apparent $V_M$ and the bead radius (Figure 4A, B). However, $K_M$ barely correlates with the bead size (Figure 4C), which explains that apparent $V_M/K_M$ toward NADPH also follows a strong negative correlation with the particle radius (Figure 4D). Likewise, $V_0$ shows a strong correlation with the radius, but the *p*-value is too high to be considered significant. That low statistical significance was mainly due to the low number of beads used for the analysis, which suggests that large sampling sizes are required to obtain reliable correlations.

Bead (or particle) size distrbution is a clear source of functional heterogeneity, but that insight cannot be generalized for all immobilized biocatalysts. Comparing Figures 3 and 4, we observe that functional variability was more noticeable for the KRED/NADPH pair than for the FDH/NAD$^+$ pair. Bead radius had a higher impact on KRED co-immobilized with NADPH (Figure 4), since the highest enzyme performance was obtained with the smallest particles. Larger particles seem to pose longer diffusion paths for TFA across the agarose microstructure, indicating that larger fractions of the immobilized KRED suffer the mass transport restrictions of the exogenous substrate. This effect was not as noticeable for FDH co-immobilized with NAD$^+$ (Figure 3), which was likely because the diffusivity of formic acid was more efficient. These insights are supported by the diffusion models based on macroscopic analysis that predict higher volumetric activities of the supported biocatalysts when enzymes are immobilized on small particles (beads) [22]. In fact, Sigurdardóttir et al. [32] have reported how the same sample of an alcohol dehydrogenase immobilized on silicon carbine microparticles decreased its intrinsic activity upon particle aggregation, as a consequence of the higher

diffusion restrictions posed by the higher size of the formed aggregates. In this study, they correlated enzyme functionality and particle size using two separated techniques, but were not able to distinguish between the apparent activities of those individual populations with different particle sizes.

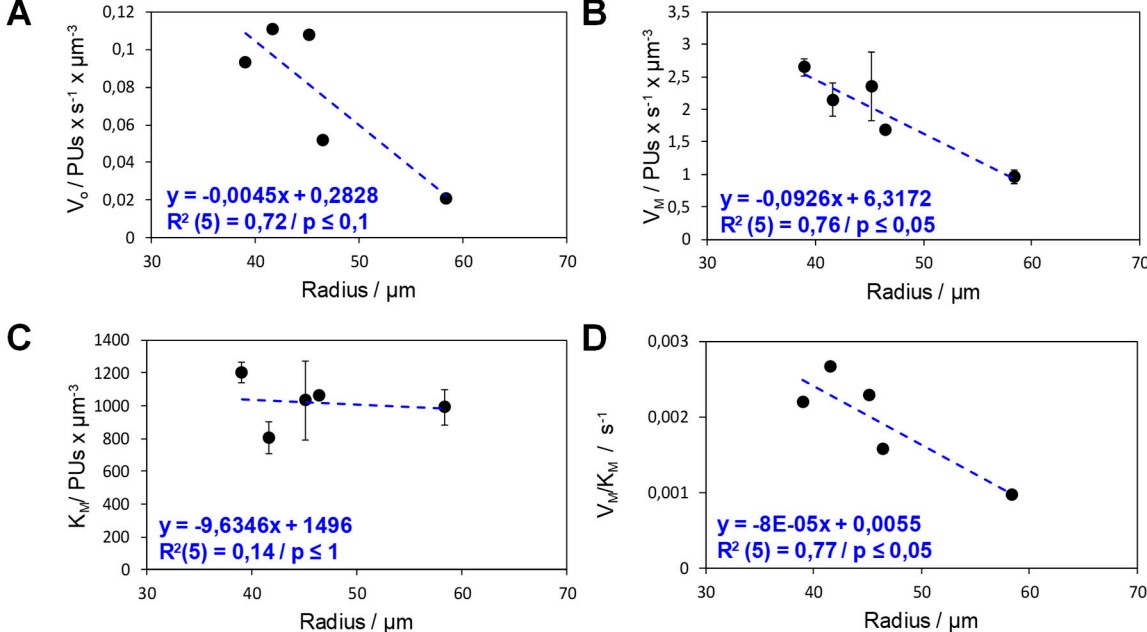

**Figure 4.** Effect of the particle radius on the kinetic parameters of the immobilized KRED and NADPH using TFA as substrate. For single particle analysis, apparent $V_0$ (**A**), $V_M$ (**B**), $K_M$ (**C**), and $V_M/K_M$ (**D**) were determined and plotted against the radius of each particle (circles). Errors for apparent $V_M$ and $K_M$ values were calculated from three different iterations of PCAT using different input data sets. The dashed blue line represents the linear correlation between each kinetic parameter and the particle radius. Linear regression equations are shown within each graph. The correlation coefficients ($R^2$) and the *p*-value were calculated with an ANOVA statistical analysis. The regression line and the statics are shown in blue. Number of samples is indicated in brackets beside the $R^2$ value.

Therefore, in operando single-particle studies merged with image analytics elicit the intrinsic sample heterogeneity of KRED co-immobilized with NADPH using TFA as the exogenous substrate, assigning functionality to each population with a specific particle size. These correlations would be impossible to prove using conventional macroscopic studies based on bulk measurements. Hence, this powerful tool provides us with the spatiotemporal resolution needed to identify the existence of bead populations with different kinetics.

## 2.3. Functional Heterogeneity of a Co-Immobilized His-BsADH/NADH Pair Using Different Carbonylic Substrates

A crude extract of *His-BsADH* was selectively purified and site-directly immobilized on agarose porous microbeads activated with cobalt chelates (AG-Co$^{2+}$). Upon immobilization, the bound enzymes were coated with PEI to enable the ionic adsorption of NADH. The resulting heterogeneous biocatalyst loaded 64 nmol of enzyme and 10 µmol of NADH per gram (Table S1). As the systems described above, this heterogeneous biocatalyst become self-sufficient upon the NADH immobilization, since no exogenous cofactor is required for the enzyme-driven reduction. Through single-particle and in operando studies, we estimated the apparent M-M kinetics of immobilized *His-BsADH* toward its co-immobilized NADH, using two different substrates: acetone and benzaldehyde.

We observed that the reduction of acetone generated functional dispersion among the analyzed beads, but that variability could not be correlated to the particle size, since the regression coefficients were ≤0.5 and the correlations were not significant ($p \le 0.5$) for all kinetic parameters herein studied

(Figure 5). On the contrary, when benzaldehyde was used as substrate, $V_0$, $V_M$, and $V_M/K_M$ followed a clear negative correlation (R ≥0.65) with the bead size (Figure 5). These results suggest that bead radius have a higher impact on the benzaldehyde than on the acetone diffusion, which points out that the benzaldehyde is less available for *His-BsADH* immobilized on larger beads than on smaller ones. This fact would explain the lower apparent $V_M/K_M$ we found within larger particles when exogenously adding the aromatic substrate. On the contrary, acetone seems to be negligibly affected by particle size, although we analyzed a wide range of bead radius (20–80 μm). These results align with those analyzed in Section 2.3 for the reduction of NADPH and oxidation of $NAD^+$ using exogenous TFA and formic acid, respectively. Hence, the effect of the bead size on the enzyme performance under aqueous conditions is more dramatic when using bulky and hydrophobic substrates (TFA and benzaldehyde) than using small and polar ones (acetone and formic acid). In porous and functionalized materials, both the polarity and size of the substrates affect their diffusion through the carriers where the active phase is immobilized [4,33]. Furthermore, the nature of the solid materials (hydrophilic or hydrophobic) can also hamper the diffusion of the substrates through their porous microstructure [14]. These diffusion restrictions tend to be more dramatic when bulky, and apolar substrates must access enzymes supported on hydrophilic agarose porous carriers. Hence, benzaldehyde seems to undergone higher diffusion restrictions than acetone across the agarose beads; however, the apparent $V_M/K_M$ of the co-immobilized His-BsADH/NADH pair is higher when exogenously adding benzaldehyde rather than acetone (Figure 5D). That better performance is due to the significantly lower apparent $K_M$ toward NADH in the presence of the aromatic substrate (Figure 5C).

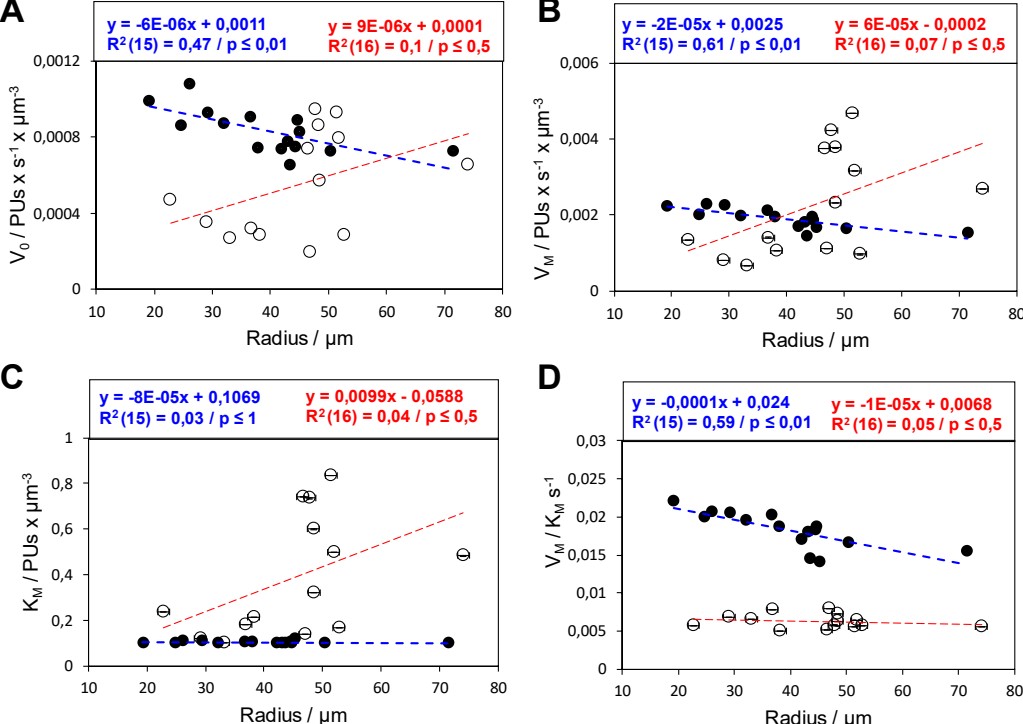

**Figure 5.** Effect of the particle radius on the kinetic parameters of the immobilized BsADH and NADH using either benzaldehyde (full circles) or acetone (empty circles) as substrate. For single particle analysis, apparent $V_0$ (**A**), $V_M$ (**B**), $K_M$ (**C**), and $V_M/K_M$ (**D**) were determined and plotted against the radius of each particle (circles). The dashed lines represent the linear correlation between each kinetic parameter and the particle radius using benzaldehyde (blue) and acetone (red) as substrates. Linear regression equations are shown in the top boxes of each graph. The multiple correlation coefficient $R^2$ and the *p*-value were calculated with an ANOVA statistical analysis and shown in blue and red for benzaldehyde and acetone, respectively. Number of samples is indicated in brackets beside the $R^2$ value.

Similar to benzaldehyde, the use of acetone as an exogenous substrate also provoked significant kinetic differences between particles, but that variability could not be attributed to particle size (Figure 5). Unfortunately, the lack of correlation between those apparent kinetic parameters and the particle size makes impossible to identify the source of functional variability according to the data herein presented. However, we found a strong correlation between the maximum formed product ([Pf]) and $V_0$ ($R^2 = 0.92$; $p \leq 0.01$) (Figure S2). [Pf] provides information about the maximum concentration of the oxidized cofactor at the end of the reaction; this parameter reflects the maximum yield reached by the reaction at the last frame of analysis. It means that beads with higher apparent enzyme activity accumulate a larger amount of product. On the contrary the correlation coefficients of [Pf] with other apparent kinetics parameters were too weak ($R^2 < 0.5$; $p \leq 0.05$). This insight corroborates that enzymes working faster accumulate more oxidized cofactors within the beads, regardless of the particle size.

Interparticle studies have enabled comparing kinetics between beads of different sizes; however, the inspection of the interior of the beads may inform about the catalytic performance of enzymes at different intraparticle regions. To that aim, we modified the previously developed image analytics routine to now collect all the information contained in one pixel (see Section 3.7.2). That tool allowed averaging the fluorescence intensity of all the pixels located at the same distance from the center of one bead. This operation generates mean progress curves that represent the average reaction time courses of pixel populations separated a certain distance from the particle center. Then, each time course was normalized, and the apparent kinetic parameters were estimated toward the immobilized NADH, as previously described for the analysis of single beads. Now, selecting the group of pixels at different distances from the center, we are able to reconstruct a functional profile that informs us about the enzyme activity at the intraparticle level. This advance in image analytics makes us gain resolution in the functionality of the immobilized enzymes. Figure 6A shows that when using acetone, apparent $V_M$ values randomly vary inside the beads, regardless of their size. On the contrary, the $V_M$ profiles with benzaldehyde showed that the NADH oxidative activity decreased from the outer to the inner regions of the particle (Figure 6B). That gradient was more drastic in small beads than in large ones. These data confirm the results obtained with the interparticle studies of NADH oxidation using benzaldehyde (Figure 5). Since the reduction of both substrates was performed with exactly the same sample with the same volumetric activity ($IU \times g_{carrier}^{-1}$), we discard the intraparticle enzyme density as the source of the variability. If that were the case, similar random variations would be observed in the $V_M$ values across the bead profile using both benzaldehyde (Figure 6B) and acetone (Figure 6A). However, irregular functional profiles only occurred when acetone was used to trigger the reduction reaction.

Both intraparticle and interparticle studies are comparable among them, but unfortunately, output data are given in arbitrary concentration units ($PU \times \mu m^{-3}$) that cannot be compared to molar units obtained for bulk studies. Moreover, we did not quantify the enzyme concentration per particle, which prevents us from comparing these kinetic data with those data reported in the literature. Despite its limitations, this method underpins our ongoing efforts to developing a more quantitative analysis where the concentration of both immobilized cofactors and enzymes can be determined using calibration methods. We are currently working on calibrating the autofluorescence of NADH and labeling the His-BsADH with compatible fluorophores to quantify their intraparticle concentrations. This quantification will allow calculating enzyme parameters such as the specific activity or $k_{cat}$, while the $K_M$ values in mM could be directly contrasted with those reported in the literature.

The calculation of these parameters is highly relevant to better understand the functional variability of those samples where there is no trend between enzyme performance and particle size. For example, beads having different $V_M$ values might be due to the diffusion restrictions related to the bead size, but also to the different enzyme loads or densities within each bead. As well as in single-cell studies [34], the fluorescent labeling of the enzyme is mandatory to decipher whether apparent kinetic parameters of the immobilized enzymes vary with the intraparticle enzyme concentration.

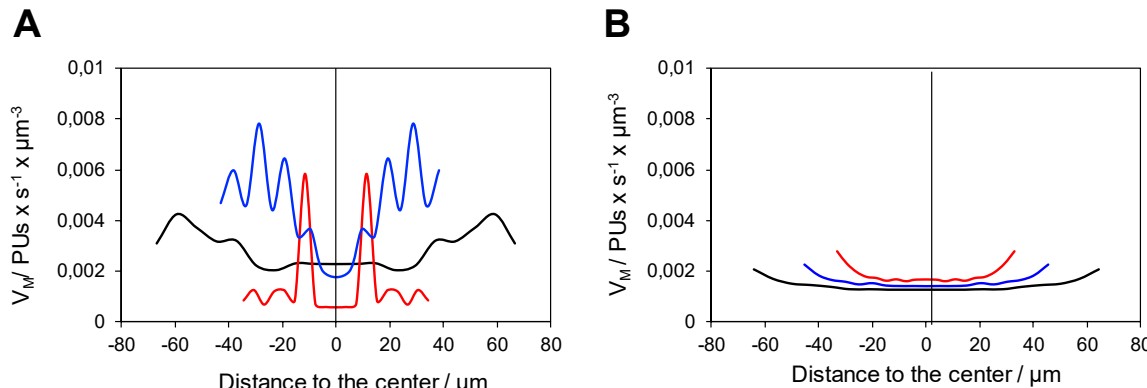

**Figure 6.** Profile of apparent $V_M$ along the radius of His-BsADH co-immobilized with either NADH (**A**) or NAD+ (**B**) using acetone and benzaldehyde as substrates, respectively. The profiles of $V_M$ were analyzed for three single beads with different radii 64–66 μm (black), 43–45 μm (blue), and 33–34 μm (red). Reactions were triggered by the addition of the 10 mM or the organic substrate (acetone or benzaldehyde) in 10 mM Tris-HCl pH 7 at 25 °C.

## 3. Materials and Methods

### 3.1. Materials

Agarose microbeads activated with cobalt chelates (AG-Co$^{2+}$) (50–150 μm diameter) were purchased from Agarose Bead Technologies (Madrid, Spain). μ-Slides VI 0.4 were purchased from Ibidi (Planegg, Germany). Nicotinamide-adenine-dinucleotide sodium salt (NAD$^+$) and nicotinamide-adenine-dinucleotide reduced sodium salt (NADH) were purchased from GERBU Biotechnik GmbH (Heidelberg, Germany). Branched polytheleneimine (PEI) 60 kDa, acetone, benzaldehyde, and other reagents and solvents of analytical grade were purchased from Sigma-Aldrich (St. Louis, IL, USA).

### 3.2. Expression and Immobilization of His-BsADH

Alcohol dehydrogenase from *Bacillus stearothermphilus* tagged with a histidine hexapeptide at its N-terminus (His-BsADH) was expressed as described elsewhere [35]. Briefly, the recombinant plasmid that harbors the gene that encodes Bs-ADH was transformed into *E. coli* BL21 (DE3) chemical competent cells. These cells were cultivated at 37 °C in LB medium containing 30 μg mL$^{-1}$ kanamycin. When the culture reached 0.6 O.D, isopropyl-1-thio-β-d-galactorpyranoside (IPTG) was added up to 1 mM to induce the expression of His-BsADH. After induction, the cells were grown for 3 h and harvested by centrifugation (10,000× *g*). Then, the cell pellet was resuspended in 25 mM of sodium phosphate buffer at pH 7, and the resulting suspension was sonicated. Afterwards, the cell debris was removed, and 5 mL of the clear crude extract were incubated with 0.5 g of agarose microbeads activated with cobalt chelates for 1 h. Then, the beads were washed with 10 mM of sodium phosphate buffer, vacuum dried, and stored at 4 °C for further use.

### 3.3. Coating of Immobilized His-BsADH with PEI

First, 0.5 g of His-BsADH immobilized on AG-Co$^{2+}$ was incubated with 5 mL of 10 mg × mL$^{-1}$ PEI 60 kDa at pH 8 for 1 h at 25 °C. Afterwards, the resin was washed 3 times with 10 volumes of 10 mM Tris-HCl at pH 7.

### 3.4. Ionic Adsorption of NAD on His-BsADH Immobilized on Ag-Co$^{2+}$ and Coated with PEI (NAD/PEI/His-BsADH@AG-Co$^{2+}$)

Then, 100 mg of His-BsADH immobilized on AG-Co$^{2+}$ and further coated with PEI was incubated with 1 mM of NADH in 10 mM of Tris-HCl buffer at pH 7 for 1 h at 25 °C. After the incubation, three

washing steps were carried out by mixing the resin with 10 volumes of 10 mM of Tris-HCl buffer at pH 7. During the whole process, the supernatants from the immobilization and the washing steps were kept and further spectrophotometrically measured at 340 nm to calculate the final amount of NADH loaded into the resin.

### 3.5. Spectrophotometric Enzyme Assay

The redox activity of His-BsADH was spectrophotometrically measured by monitoring the absorbance at 340 nm, which varied depending on the concomitant production or consumption of NADH. The reaction was carried out at 25 °C and pH 7; then, 190 μL of 100 mM acetone and 0.1 mM of NADH in 25 mM of sodium phosphate buffer were incubated with 10 μL of enzymatic solution (either soluble or immobilized). One unit of activity was defined as the amount of enzyme that was needed to either reduce or oxidize 1 μmol of the corresponding nicotinamide cofactor per minute at 340 nm, 25 °C, and pH 7.

### 3.6. In Operando Activity Assays through Time-Lapse Fluorescence Microscopy

Different redox reactions were performed under the fluorescence microscope using a channel slide (μ-Slide VI 0.4). First, 170 μL of 1:85 (*w/v*) suspension of the self-sufficient heterogeneous biocatalyst (10 mg $_{\text{His-BsADH}}$ and 10 $\mu mol_{\text{NADH}}/g_{\text{carrier}}$) in 10 mM of Tris-HCl at pH 7 was placed into the channels. Reactions were triggered with 10 μL of 1 M of either acetone or benzaldehyde dissolved in acetonitrile. The final reaction mixture contained 55 mM of substrate and 5.5% acetonitrile. As the control experiment, in operando reactions were triggered with 10 μL of pure acetonitrile. Under the microscope, the NADH fluorescence intensity was recorded every 6 s using an Axio Obserber Zeiss epifluorescence microscope with a Colibri LED illumination module using a 365-nm LED and an emission filter of 420–470 nm using a beamsplitter at 395 nm. The sample was observed with an EC Plan Neofluar 10X objective with a numerical aperture of 0.30 and coupled to an apotome grid VL with a working distance of 5.3 mm. The brightfield channel was also recorded to detect any change in the bead positions that might create an artifact for further analysis. Images were taken and recorded with an AxioCam MRm (Zeiss, Oberkochen, Germany) with 1388 × 1040 resolution.

### 3.7. Image Processing and Analytics

3.7.1. Identification of Regions of Interest (ROIs) through Image Segmentation

To identify each bead for further single-particle analysis, an Image J plugin (Plugin1) was written to select each particle as an individual ROI based on the intensity differences between the fluorescence inside and outside (background) the particle. This plugin stacks all the temporal images of one specific ROI creating a file only including the fluorescence channel for further analysis. The challenge of this segmentation is selectively identifying those ROIs that keep the same position along the time-lapse experiment. To validate the efficiency of the automatic selection, the selected ROIs were cured by comparison with the original brightfield images to assure that the bead contour did not exit the ROI during the whole experiment. To finally select the ROIs, two additional criteria were introduced into the plugin: circularity and isolation. Since the carrier particles are perfectly defined spheres, all the ROIs must keep a circularity value above a threshold annotated into the script. The circularity value can be changed based on the nature of the sample to endow the plugin with analytical flexibility. The other criterion was isolation, in order to discard those beads either touching the edges of the image or the border of neighbor beads.

3.7.2. Obtaining Reaction Progress Curves of Single Particles

To this aim, a new Image J plugin (plugin 2) was written to obtain time-course plots from files created in 3.7.1. This plugin calculates the average fluorescence intensity (RFU) within the ROI for each time and plots that charge the intensity value versus time. To increase the spatial resolution

of the analysis, a new version of this plugin (plugin 2.p) was created to obtain reaction progress curves pixel by pixel. However, both plugins (for particles and for pixels) provide raw values of fluorescence intensity that are hardly comparable between different particles from the same sample and from samples measured in different experiments. To further ease the kinetic analysis and comparison between particles and samples, we introduced a series of normalization steps.

(i) Time courses must follow a growth path for further kinetic fitting. For those reactions where the fluorescence decays along the time, the fluorescence intensity values within each bead are converted into arbitrary product units (*PU*), which provide an estimation of the product formed during the enzymatic reaction (i.e., reduction of NADH to NAD$^+$). To estimate the *PU* values at any time (*n*), the raw data from the time courses were inverted using the following (Equation (1)):

$$PU_{t\,=\,n} \;=\; RFU_{t\,=\,0} - RFU_{t\,=\,n} \tag{1}$$

where that $RFU_{t\,=\,0}$ means the average fluorescence intensity at time 0 corresponding to one ROI or pixel, and $RFU_{t\,=\,n}$ means the average fluorescence intensity at one specific time corresponding to the same ROI or pixel. According to this normalization, the reaction progress curves now follow a growth path with a starting coordinate (X,Y) of (0,0). In the case of reactions where the fluorescence intensity increases along the time, this normalization is not required, but the $RFU_{t\,=\,0}$ must be subtracted from all the data points to achieve $PU_{t=0} = 0$.

(ii) Time courses must not contain lag phases for further kinetic analysis. For each time course, the lag phase was removed by assigning t = 0 to the time where product units present a value of 15% of the maximum product units of the analyzed time course. Then, the newly assigned $PU_{t=0}$ value is subtracted from all the data to make a plot with starting coordinate (X,Y) of (0,0) (see Figure S3).

(iii) Normalized time courses must not contain outlier values to enable the subsequent kinetic fitting. In those cases, where some experimental artifacts (i.e., an unexpected illumination of the work zone during the time-lapse experiment) occur, the affected data must be removed (see Figure S4).

(iv) The arbitrary product units must be corrected by bead volume (see Figure S5). Since 3D microbeads are the objects of analysis and the data were acquired with an epifluorescence microscope, the captured intensity values correspond to all photons emitted by the focal volume. This means that the signal acquired for one bead depends on the bead volume (Figure S6). Hence, a volume correction is needed for each data point to compare kinetics among beads. The volume correction was applied using Equations (2) and (3):

$$Bead\,volume\;[\mathrm{\mu m}^3] \;=\; \frac{4}{3}\pi \times r_p^3 \tag{2}$$

where *r* is the radius of the bead.

$$Bead\,volumetric\,Intensity\;[\mathrm{PU}_{t\,=\,n} \times \mathrm{\mu m}^{-3}] \;=\; \frac{Total\ Intensity\ per\ bead\ [\mathrm{PU}_{t\,=\,n}]}{bead\,volume\ [\mathrm{\mu m}^3]} \tag{3}$$

The volume correction at the pixel level is slightly more complex than at the particle level, as the pixel coordinates must be considered to calculate the focal volume. Pixels closer to the bead center correspond to larger focal volumes than pixels positioned further from the center (Figure S5). Accordingly, the fluorescence intensity of each pixel was corrected for the specific volume of that pixel ($V_p$), considering each pixel as a rectangular prism, whose area is the pixel resolution given by the microscopic image ($A_p$) and the height (*h*) is given by the root square of the radius of the particle (*r*) minus the distance (*d*) between the pixel position and the bead center (Figure S5), multiplied by 2. The volumetric correction for the average intensity of all pixels located at the same distance from the bead center was done according to Equations (4) and (5):

$$Vp\;[\mathrm{\mu m}^3] \;=\; A_p \times 2h \;=\; A_p \times 2\,\sqrt{r^2 - d^2} \tag{4}$$

$$Pixel\ volumetric\ intensity\ [\mathrm{PU_{t\,=\,n}}\ \mathrm{x}\ \mu m^{-3}]\ =\ \frac{Intensity\ per\ pixel\ [\mathrm{PU_{t\,=\,n}}]}{Vp\ [\mu m^3]} \tag{5}$$

All the normalization steps were done using the raw data exported from Image J and imported to Excel 2010 software. As output, files containing the normalized data were created and further used for mathematical fitting.

### 3.8. Fitting of Normalized Time Courses to Determine the Kinetic Parameters of the Immobilized Enzymes

Using the Excel sheet format generated as described in Section 3.7 as the input file, the normalized time courses were analyzed with a MATLAB-based tool (PCAT) developed by Bäuerle et al. [30]. Briefly, the program fits the experimental data to the Lambert W function of the Michaelis–Menten equation (Equation (6)):

$$[S](t)\ =\ K_M W\!\left(\frac{[S_0]}{K_M}\exp\!\left(\frac{[S_0]-V_{max}t}{K_M}\right)\right) \tag{6}$$

where $W$ describes the inverse relation of the function $f(z) = z \ x \ e^z$. The time courses were fitted using an analytical model where the output data depend on input arbitrary values for $K_M$ and $V_M$. Hence, three fitting iterations using three different input data sets were performed to accurately calculate the output values. For each time course, a mean value of the Michaelis–Menten constant ($K_M$), maximum velocity ($V_M$), and catalytic efficiency ($V_M/K_M$) toward the immobilized cofactor and maximum $NAD^+$ (product) concentration ($[P_f]$) were calculated with their corresponding standard deviations using the output values resulting from the three iterations. The convergence between the three interactions is given by the standard deviation obtained for each parameter estimated toward the immobilized cofactor by PCAT.

### 3.9. Initial Rate Analysis

To calculate the initial rate from each progress curve, data with PU values lower than 20% of the maximum product concentration ($[P_f]$) were fitted using a liner regression ($[PU/\mu m3] = V_o \times t$), where the slope is the initial rate of the enzymatic reaction ($V_0$).

### 3.10. Statistical Analysis

To assess the confidence of the results herein presented, we carried out an ANOVA analysis that provides a correlation coefficient ($R^2$) and the *p*-value for each set of data. The $R^2$ means the strength of the correlation between the different estimated parameters and the particle size. When $R^2$ is close to 1, it indicates the certainty of the correlation, which can be higher or lower depending on the regression constant (b) in the linear regression equation y = a + b*x. The *p*-value indicates the significance of the differences between the plotted data. This value reflects the probability of the null hypothesis (apparent parameters are equal for all the beads). If the critical value is equal or lower than the standardized threshold $p = 0.05$, the null hypothesis is discarded; therefore, we can assure that the apparent kinetic parameters change with the bead radius

## 4. Conclusions

In this work, we have developed an image analytical tool merged with a MATLAB-based mathematical analysis that allows quantifying the apparent kinetic parameters of alcohol dehydrogenases co-immobilized with $NAD(P)^+/H$. This tool enables single-particle analysis both at bead and pixel levels that reveal the functional heterogeneity of the tested immobilized enzymes. We found that one of the sources for that heterogeneity (variability of apparent kinetic parameters) is the size polydispersity of the carrier samples. For some specific cases, we found a significant negative linear regression between enzyme catalytic performance and bead size. This size dispersion is a feature underlying the most of the commercially available carrier materials for enzyme immobilization. Hence, tools such as the one herein developed will contribute to identify technical bottlenecks in

the fabrication of readily heterogeneous biocatalysts. Through measuring the kinetic parameters of enzymes confined into a porous environment, we are approaching intracellular working conditions (100–300 mg $\times$ mL$^{-1}$) [36]. This analytical tool measures enzymes working in a packed environment, but without the biochemical noise of single-cell studies. Thus, we envision that these single-particle analyses will provide fundamental understanding about enzyme functionality under dense and crowded conditions. In order to extract more information from these image-based microscopic studies, we forecast the development of new strategies that link functional and structural variability at the protein–solid interface under operando conditions. This idea has begun to be explored in the last years using model solid surfaces [37], but direct characterization of "ready-to-work" materials would have a much higher impact on enzyme immobilization for industrial purposes.

**Supplementary Materials:** The following are available online at http://www.mdpi.com/2073-4344/9/11/896/s1. Table S1. Immobilization parameters of His-BsADH co-immobilized with NADH on agarose microbeads 9 activated with cobalt chelates; Figure S1. Effect of the maximum intraparticle cofactor concentration [Pf] on the kinetic parameters of the 11 immobilized; Figure S2. Example of lag-phase removal for one single-particle experiment; Figure S3. Removal of outlier points from one single-particle experiment; Figure S4. Calculation of volumetric fluorescence intensity; Figure S5. Graphical explanation of how to calculate the volumetric fluorescence intensity; Figure S6. Cofactor distribution across the surface of agarose microbeads functionalized with PEI.

**Author Contributions:** F.L.-G. conceived the experiments and E.M.-M. analyzed the data. E.M.-M., S.V.-L. and A.I.B.-M., performed the experiments. M.J.M. and P.R.-C. developed the ImageJ scripts. F.L.-G. wrote the paper. All authors contributed and approved the final version of the manuscript.

**Funding:** A.I.B.-M. and F.L.-G. are grateful to MINECO (BIO2015-69887-R, BIO2014-61838-EXP and PCI2018-092984) and HOMBIOCAT ERA-CoBioTech project for funding their research. We also thank ARAID and IKERBASQUE foundations for funding F.L.-G. and P.R.-C., and S.V.-L. thanks the Mexican Council of Science and Technology (CONACyT) for the postdoctoral fellowship she received.

**Acknowledgments:** Authors would like to acknowledge the use of Servicio General de Apoyo a la Investigación-SAI, Universidad de Zaragoza. We also thank to Sánchez-Ruiloba for her help and advice to analyze the fluorescence microscopy data.

**Conflicts of Interest:** The authors declare no conflict of interest.

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
