# Peer review of "Deciphering the Effect of Microbead Size Distribution on the Kinetics of Heterogeneous Biocatalysts through Single-Particle Analysis Based on Fluorescence Microscopy"

_catalysts, doi:10.3390/catal9110896_

Round 1
Reviewer 1 Report
Review catalysts –620450 for the Authors
This is a very interesting work on an important aspect, namely the optimization of heterogeneous biocatalysts for industrial purposes. The image analysis tool developed by the Authors combined with mathematical analysis based on MATLAB can contribute to the identification of technical bottlenecks in the production of readily heterogeneous biocatalyst. The Authors also forecast the development of new strategies combining functional and structural variability at the protein-solid interface under operando conditions. The topic remains open for further analysis.
Carefully described individual stages of the analysis, additionally supported by illustrations and charts, emphasize the value of this work.
Author Response
We appreciate the comments of Reviewer 1 about our work.
Reviewer 2 Report
I have some comments to the manuscript:
Line 60: There is no reference on Table 1 in the text. It should be added before the Table.
Does Table 1 represent information on silicagels? Why were silicagels used to compare with agarose microparticles of this work?
In comments to Lines 68-69 (“Unfortunately, these studies mask the functional differences between populations of beads with different sizes”) and Lines 100-101 (“These data confirm the macroscopic studies[20-22,29] that demonstrate the effect of the particle size on the catalytic effectiveness”) I need to note, that it is necessary to discuss not only the SIZE of particles, but also the NATURE of the beads used for enzyme immobilization, whereas authors pay attention only to SIZE.
Special note: It is necessary to precisely articulate a goal of the work at the end of the Introduction. Does it look like: ”To quantitatively characterize the kinetics of “ready-to-use” alcohol dehydrogenases co-immobilized with their corresponding cofactors to enhance the cost-efficiency of the process”? Or is the purpose following: “Development of new strategies that link functional and structural variability at the protein-solid interface under operando conditions?” The conclusion should be close to the initial purpose and reflect its achieving.
“Results” section is rather “Results and Introduction and Discussion”. According to Template of the Journal, Results and Discussion should be given separately.
Lines 108-140: This text should be shortened or redirected to Introduction since it was published previously (Refs.27,28), wasn’t it?
I consider that RFUs and PUs may be useful for the domestic/internal use but are very incomprehensible and even shocking for the readers. Authors should maximize reader-friendless and replace these artificial units with more usual concentrations within the text and figures.
Tables within Fig.2 have strange units. Also, correlation coefficient and chi-square values for approximations should be presented. Why were three iterations implemented for approximation (line 175)? What were the convergence and/or chi-square criterion and/or something?
Line 188: “Although both V0 and VM showed a slight negative correlation with the radius size”. Here and further such negligible trend is rather undistinguishable from the constant level. If authors will reveal an equation of such line and coefficients with errors, everyone could see and conclude. So, authors should give more weighty arguments or correct these statements everywhere in the text.
Also, while the radius was increased, enzyme concentration/quantity was increased simultaneously. It could lead to improvement of total activity. Why was it not the case? It seems that there are limitations to: a) fluorescence penetration from the inner core; b) diffusion of substrate inside microparticle; c) overswing (i.e. fluorescence intensity from the pixel is so much, that used microscope sensitivity is not enough, and/or that further increase of intensity couldn’t be detected); d) surface functionality; e) affinity of the carrier to substrate/product/co-enzyme; f) etc. These versions may be discriminated if there will be quantifications, i.e. concentrations and additional information about device, etc.
Lines 197, 230, 250, 278, etc.: What are the “correlation coefficient” and “p” values determined by ANOVA? How were they calculated? Why were “p” values “less or equal” (≤) to something? What were the real values?
Lines 208-210: “That sample heterogeneity differed between the two tested supported enzyme/cofactor pairs although the kinetic parameters were estimated in the same order of magnitude for both systems.” Where are these data? Where is analysis of size distribution? Why did differing enzymes have activity of “the same order of magnitude”?
Lines 213-214: “[That] larger fractions of the immobilized KRED suffer the mass transport restrictions of the exogenous substrate.” If there were mass transport restrictions, it would be seen as apparent decrease of substrate concentration and subsequent increase of Km. However, this was not the case.
Lines 232-233: “elicit the intrinsic sample heterogeneity”. No. Please, see above.
Lines 252-253: “These results suggest that bead radius have a higher impact on the benzaldehyde than on the acetone diffusion”. No. It is rather a sign that something happens with microparticles or enzyme or both. Another evidence is the widest distribution of catalytic characteristics of enzyme immobilized on ca. 100 μm microparticles.
Figure 5d: There is wrong caption of axis (VM/ PUsx is in Picture, and VM/KM is in capture).
It will be wonderful to compare catalytic characteristics of immobilized enzymes with the native ones used in the work.
Lines 266-271: So, it could be concluded that diffusion of bulky substrate is insignificant factor for this catalytic system.
“resulting in beads with higher and lower NADH loads” (line 291), and “enzyme loading may be another source of functional variability” (lines 294-295) It is another reason to quantify them. Absolute fluorescence? Elution and analysis? Conjugated kinetic analysis? etc. Also, quantity of co-enzyme was not varied, but it has its own Km and could be measured.
Lines 295-296: “enzyme density would vary from particle to particle” Absolutely no! If there was a good mixing and enough excess during immobilization, enzyme should be distributed homogenously.
Line 298: “enzyme concentration per bead is impossible to quantify”. No. Enzyme could be eluted (imidazole, EDTA, etc.) or digested, or tagged, etc. Authors could investigate a distribution/quantity of microparticles and calculate enzyme quantity per particle. It is also necessary to evaluate the ‘true’ kinetic parameter (Vm/E0) rather than useless Vm.
Lines 316-321: This raises the question about enzyme elution/denaturation/etc., or microparticle structure.
Line 358: “Isopropyl” is missed in 1-thio-β-d-galactorpyranoside (IPTG).
Line 360: “centrifugation (1699 rcf)” The g value is preferable.
Lines 382-383: “One unit of activity was defined as the amount of enzyme that was needed to either reduced or oxidized 1 μmol of the corresponding nicotinamide cofactor at 340 nm, 25 ºC and pH 7.” What was the time used to calculate the activity: for 1 min?
Lines 397-409: Additional information is required about: a) spatial resolution; b) intensity scale; c) color range (widening the emission peak); d) etc.
Line 437 and further: “the arbitrary product units must be corrected by bead volume”. This is not evident since these light sources are spherical. Intensity of visible pixel is proportional to the total surface (4πr2) and not to the volume (4/3πr3). However, it will increase on the edge non-linearly. So, all parameters should be recalculated.
Now, Eq(6) has no any “t” (time) variable under exponent, but it should be (See, Biochemical Engineering Journal 63 (2012) 116–123). The equation should be checked.
Figure S1. The end of title should be added (see in bold): Effect of the maximum intraparticle cofactor concentration on the kinetic parameters of the immobilized His-BsADH and NADH using acetone as substrate.
Figure S4. Some words are missing at the beginning of the title (…of volumetric fluorescence intensity) that should be added.
References: Please, include the digital object identifier (DOI) for all references where available. Now DOI is absent in all references.
Author Response
We appreciate the comments of Reviewer 1 about our work.
Please see attachment

Reviewer 3 Report
What is the significance of the bold standard (50-150) for ABT in Table 1?
Author Response
What is the significance of the bold standard (50-150) for ABT in Table 1?
The size dispersion of ABT agarose beads should not be bolded, this is a mistake that has been corrected in the revised version of the manuscript. We apologize for this mistake that may drive the reviewer to missunderstand the table description.
Round 2
Reviewer 2 Report
We used three interactions because the authors that developed the PCAT stated that using the analytical mode, the outcome values depend on the input VM and KM values before starting the analysis. For this reason, we iterate the experimental data with different input values for each constant in order to obtain reliable apparent parameters. The convergence between the three interactions is given by the standard deviation annotated to the right of the mean value of each parameter estimated by PCAT (Pf, Km, Vm and Vm/Km)
It should be added to the Materials and Methods section.
In the ANOVA test, we compared whether the apparent kinetic parameters vary with the particle size, and if those variations were significant. To assess the significance of the differences among the beads with different radius, we calculate the p value that means the probability of the null hypothesis, in this case that apparent parameters are equal for all the beads. If the critical value is below the standardized threshold p=0.05, the null hyphothesis is discarded, therefore we can assure that the apparent kinetic parameters change with the bead radius. Normally, p values are given in intervals since the absolute value is meaningless. In this statistic study, we focused on assessing whether the correlation between the kinetic parameters and the bead radius were significant or not.
It should be added to the Materials and Methods section.
Unfortunately, we do not understand what the reviewer 2 means in this comment.
It means that heterogeneity was not issued (like in lines 208-210), and thus there is no proofs to the statement.
The clearest evidence that support that fact, it is the dependence of the apparent maximum rate on the particle radius only when bulky molecules are used as exogenous substrate.
No, since all established parameters are questionable now (see last comment).
In fact, this is what we have done in this work, we have quantified the KM, among other parameters, of the immobilized enzymes towards their immobilized cofactors. KM values of immobilized cofactor are different from the values found in solution as we have already reported in Velasco-Lozano et al Angewandte, 2017, 56, 771-775, using bulk studies.
It is necessary to emphasize which kinetic parameters were measured towards co-factors. Anyway, I misunderstand how is it possible since authors assure that it is impossible to enumerate a co-factor quantity within microparticles.
We agree with reviewer 2, but only under the conditions she/he describes; it means under loadings that saturate the carrier surface area. However, in the vast majority of immobilization protocols, the enzymes are immobilized in much lower concentration than the maximum capacity of the carrier. In this scenario, we have observed that the density and the distribution can be controlled. Furthermore, we have even found significant heterogeneity within the same sample (see results for the KRED shown above). We know that depending on the immobilization kinetics both the enzyme density and its position across the microstructure of porous materials may vary (see reference Bolivar et al, 2011, Journal of biotechnology 155, 412-420)
In this way it raises the questions of reproducibility. May be, it will be better to avoid such suggestion?
According to the information we acquired from the epifluorescence microscope, we analyzed the total intensity in our 2D ROI; this is the 2D projection of a 3D object. This is based on the assumption that agarose presents a similar refraction index as water and discarding any optical artifacts related to the objective. Therefore, pixels at the edges will accumulate less emitted photons (lower number of pixel stacked at that XY coordinate) because the height of the bead at that position tends to 0 (see the figure below). On the contrary, the highest amount of photons will be accumulated at the center of the beads (higher number of pixels stacked at that XY coordinate). Since all the beads are assumed to have the same cofactor density, we need to account for total intensity in our volume.
Refraction has nothing to do with absorption. The more height will cause more (inner) absorption. So, we have returned to the question of enzyme/co-factor/substrate distribution inside the particle…
When we do the calculations using the total fluorescence intensity per area, we still observe that RFU/um2 increases along the particle size at the frame time=0. This linear correlation masks the relationship between the enzyme activity and the particle size. Fortunately, we removed the linear correlation between the initial intensity and the particle size by normalizing the total intensity per bead regarding to the volume of each bead. The figure below this text shows the different linear regression coefficient depending on the normalization done. The lowest correlation between initial volumetric intensity and the particle size is the reson why we chose the volumetric fluorescent intensity to obtain the reaction time courses and perform the kinetic analysis at both particle and pixel level using images form epifluorescence microscopes. If we had developed this method using data from a confocal microscope where pinhole had been similar to the pixel resolution, we could have expressed the product units as intensity per area because the contribution of the Z-dimension to the 2D image would have only been 1 pixel. Figure S5 intends to graphically explain this correction, we have added part of the information shown below.
This is perfect! I can illustrate: here I have entered a circle in your profile (with blue); further I widened it (to fit the profile) and enlarged in two times (as you suggested to ‘account for a volume’, i.e. intensity has doubled due to height). As a result, profile is poorly fitted by this interpretation (red curve). However, half of the ellipsis (pink) with ellipticity of ca. 0.53 being enlarged in two times perfectly fits!
That’s it; you have excessively taken into account the particle radii (when relating RFU to particle volume). Further, you have overestimated all the kinetic parameters and obtained questionable ‘decreasing trends’ (while the particle size was increased). It was wrong. So, I insist on you should recalculate all kinetic parameters or, at least, do not use such over-normalization. You have lost the true trend.
(please see the picture in attached file).
